# Disease Perception and Coping with Emotional Distress During COVID-19 Pandemic: A Survey Among Medical Staff

**DOI:** 10.3390/ijerph17134899

**Published:** 2020-07-07

**Authors:** Milena Adina Man, Claudia Toma, Nicoleta Stefania Motoc, Octavia Luiza Necrelescu, Cosmina Ioana Bondor, Ana Florica Chis, Andrei Lesan, Carmen Monica Pop, Doina Adina Todea, Elena Dantes, Ruxandra Puiu, Ruxandra-Mioara Rajnoveanu

**Affiliations:** 1Department of Medical Sciences—Pulmonology, Faculty of Medicine, Iuliu Hatieganu University of Medicine and Pharmacy, 400012 Cluj-Napoca, Romania; manmilena50@yahoo.com (M.A.M.); anna_f_rebrean@yahoo.com (A.F.C.); andrei_lesan@yahoo.com (A.L.); cpop@umfcluj.ro (C.M.P.); doina_adina@yahoo.com (D.A.T.); andra_redro@yahoo.com (R.-M.R.); 2“Leon Daniello” Clinical Hospital of Pulmonology, 400332 Cluj-Napoca, Romania; viana115@yahoo.ca (O.L.N.); ruxi.puiu@yahoo.com (R.P.); 3Faculty of Medicine, Carol Davila University of Medicine and Pharmacy, 020021 Bucharest, Romania; claudiatoma@yahoo.co.uk; 4Department of Biostatistics, Iuliu Hatieganu University of Medicine and Pharmacy, 400012 Cluj-Napoca, Romania; cosmina_ioana@yahoo.com; 5Faculty of Medicine, Ovidius University, 900740 Constanta, Romania; elena.dantes@gmail.com

**Keywords:** COVID-19, emotional distress, stress perception, cognitive coping

## Abstract

The novel coronavirus disease, COVID-19, is a highly contagious infectious disease declared by the World Health Organization to be a pandemic and a global public health emergency. During outbreaks, health care workers are submitted to an enormous emotional burden as they must balance the fundamental “duty to treat” with their parallel duties to family and loved ones. The aims of our study were to evaluate disease perceptions, levels of stress, emotional distress, and coping strategies among medical staff (COVID-19 versus non-COVID-19 departments) in a tertiary pulmonology teaching hospital in the first month after the outbreak of COVID-19. One hundred and fifteen health care workers completed four validated questionnaires (the brief illness perception questionnaire, perceived stress scale, the profile of emotional distress emotional, and the cognitive coping evaluation questionnaire) that were afterwards interpreted by one psychologist. There was a high level of stress and psychological distress among health care workers in the first month after the pandemic outbreak. Interestingly, there were no differences between persons that worked in COVID-19 departments versus those working in non-COVID-19 departments. Disease perceptions and coping mechanisms were similar in the two groups. As coping mechanisms, refocusing on planning and positive reappraisal were used more than in the general population. There is no difference in disease perceptions, levels of stress, emotional distress, and coping strategies in medical staff handling COVID-19 patients versus those staff who were not handling COVID-19 patients in the first month after the pandemic outbreak.

## 1. Introduction

The outbreak of COVID-19 (coronavirus disease 2019), caused by a new coronavirus known as Severe Acute Respiratory Syndrome Coronavirus 2 (SARS-CoV-2), was discovered for the first time in December 2019 in Wuhan (China) and spread rapidly in almost all regions of the globe. The World Health Organization (WHO) declared in March the COVID-19 pandemic a global public health emergency [1]. The history of humanity has been marked by many fearsome epidemics of infectious diseases (Ebola, Zika, and Spanish Flu). In the last few years, the international community has experienced a similarly frightening public health emergency on a global scale, with the spread of severe acute respiratory syndrome (SARS) [2]. Because of globalization, an infectious disease can travel from one continent to another in a matter of hours and can increase the importance of the international coordination of efforts to respond to new outbreaks of infectious disease [1,2]. The novel coronavirus disease is called COVID-19 [3]. Given the contagious nature and the prolonged incubation period of COVID-19 infection (2 up to 14 days, with an average of 5 to 6 days), many health care workers may have feared that they could infect their family members, friends and colleagues [3,4]. Wu and colleagues showed, in 2009, during the SARS epidemic, the necessity of understanding the possible psychosocial impacts of an outbreak with an easily transmitted, rapidly spreading infectious disease among health care professionals [5]. In the previous pandemic, it was emphasized that health care professionals were under intense stress from fear of becoming ill, spreading the infection to their families, and the heavy workload [2,6,7,8]. During outbreaks, health care providers must balance the fundamental “duty to treat” with their parallel duties to family and loved ones. Health caretakers were forced to deal with the loss of life while processing their survival in the setting of significantly decreased social support [7]. Studies of the last SARS outbreak found the enormous emotional burden for those health care workers who were on the first lines of the battle against the disease [2,4]. In addition to physical stress, medical staff also face huge mental burdens [9]. Even people with no comorbidities can contract the disease and become critically ill. In the current COVID-19 pandemic, health professionals are confronted with specific stressors and risks for physical and mental health [10]. During the influenza pandemic in 2009–2010, it was seen that risk perceptions influence individual protective behaviors. The risk perceptions are not necessarily correlated with the actual risk and perceived exaggeration was associated with a reduced probability to implement the recommended practices [11]. In the current COVID-19 pandemic, health workers from all professions are facing significant challenges in coping with the crisis [10]. Coping with the crisis is a real challenge, especially because people coping strategies will also change in this kind of situation. Coping represents the thoughts and actions that individuals use to deal with stressful events. There are two general coping strategies: problem-focused coping, where the purpose is to solve the problem or change the situation, and emotion-focused coping, which aims to reduce the emotional distress associated with stressful situations [9,10]. Different emotions led to various coping strategies. For instance, in persons who report anger and fear, active-oriented coping strategies such as searching for information and asking questions, are frequently used. The who are sad are more likely to use non-active coping strategies such as avoiding or accepting problems. Nevertheless, coping strategies are essential, successful use of coping strategies will help individuals manage stressful events and reduce negative emotions [10]. The mental health care for health professionals directly affected by the COVID-19 epidemic has been under-addressed. The present study aims are to evaluate disease perceptions, level of stress, emotional distress, and coping strategies for dealing with the COVID-19 pandemic among healthcare workers from pulmonology teaching hospital from Cluj-Napoca, Romania, in the first month after the outbreak started. COVID-19 staff worked in the high-risk areas of the hospital where they were continuously exposed to patients with COVID-19, and the answers were compared with the team that worked in the same pulmonology hospital, in the non-COVID-19 department. We hypothesized that those working in the COVID-19 department would be more stressed and more emotionally distressed than the non-COVID-19 department staff. A subgroup analysis was made in the COVID-19 group: intensive care unit staff versus pulmonology ward staff, as the intensive care unit contains more severely ill patients. 

## 2. Materials and Methods

A cross-sectional study consists of a survey of the employees working in a pulmonology hospital dealing with COVID-19 patients. The hospital, situated in one of the biggest cities in the western part of the country, was in the first line in the battle against COVID-19. It’s a teaching hospital with two departments, non-COVID-19 (dealing with respiratory diseases others than COVID-19), and a COVID-19 department that comprises a pulmonology ward and an intensive care unit department. 

### 2.1. Participants

All health care workers from the hospital were involved (doctors, junior doctors/residents, nurses, and caregivers). Work time was 8 h shift/and 24 h break without any free days (no weekend, no holidays) for the COVID-19 department and eight hours shifts, with free weekend days for the non-COVID-19 participants, but with the assurance of health care equipment protection for all participants. The study examined the hospital employees disease perceptions, level of stress, emotional distress, and the strategies used for coping with the crisis using a validated questionnaire. The study instrument consisted of four questionnaires: the brief illness perception questionnaire (IPQ), a perceived stress scale (PSS 10), a profile of emotional distress (PDE), and a cognitive-emotional coping questionnaire (CERQ). The questionnaires were completed by each participant individually in the following order: PSS-10, PDE, IPQ, and CERQ. CERQ would be the last questionnaire to be completed. It’s essential to evaluate the stress and emotional distress first and then to see how you can cope with them. 

### 2.2. The Brief Illness Perception Questionnaire (IPQ)

This is a nine-items questionnaire elaborated by Elisabeth Broadbent and colleagues [12] that assessed cognitive perceptions of disease in terms of: effect on life—consequences score (IPQ 1); duration of illness—timeline score (IPQ 2); control over illness—personal control score (IPQ 3); beliefs about the effectiveness of treatment—treatment control score (IPQ 4); experience of symptoms—identity score (IPQ 5); concern about illness—illness concern score (IPQ 6); and degree of understanding of the disease—coherence score (IPQ 7). IPQ 6 and IPQ 8 reflect a combination of emotional and cognitive representation, question 8 being a multifaceted question about mood. Question number 9 (IPQ 9) is the causal item, evaluating the three most important factors causing their illness. The responses given by the participants in our study were grouped into six categories as follows: 1—civic responsibility, 2—epidemiological context, 3—Incompliance with work rules, 4—preexisting medical conditions, 5—at-risk behaviors, 6—physiological factors [12,13]. Each question assessed one dimension of illness perceptions by giving a score from 0 to 10, 0 being least affected, 10 being highly affected, and, therefore, having a more catastrophic perception of the disease. Scores therefore varied between 0, the lowest, and 80, the highest. 

### 2.3. Perceived Stress Scale (PSS 10)

The PSS 10 is a multiple-choice questionnaire, assessing the level of stress perceived in the last two weeks. It determines to what extent one person sees aspects of his life as being uncontrollable, unpredictable, and demanding. It has ten questions, and the responses are on a Likert scale of 5 points, from 0 (never) to 4 (very often). The scores vary between 0 and 40, the higher the score, the higher the level of perceived stress. 

A score between 0 and 12 represents a medium-low level of stress. This result has no clinical significance. It indicates that in recent weeks, the person has not been exposed to many demanding situations or has managed to cope with them without great difficulty; it is characteristic of the average population. A score between 13 and 19 represents a medium-high level of stress and may have clinical significance. The result indicates that in recent weeks one person has been exposed to many demanding situations, which they have faced with difficulty. It is characteristic of people with stress problems. A score over 20 represents a high level of stress and may have clinical significance. It means that in recent weeks the person has faced many demanding situations and has not been able to cope with them or has overcome them with great difficulty. It is characteristic of people with significant stress problems [14]. 

### 2.4. Profile of Emotional Distress (PDE)

It is a questionnaire that measures the presence of negative emotions, both functional and dysfunctional (such as anxiety, worry, depression, sadness). It comprises 26 adjectives that describe negative emotions that are grouped into 6 subscales:6 items represent functional negative emotions in the “sadness/depression” category.8 items represent dysfunctional negative emotions in the “sadness/depression” category.6 items represent functional negative emotions in the “fear” category.6 items represent dysfunctional negative emotions in the “fear” category.12 items represent functional negative emotions ((sadness/depression) and “fear”).14 items represent dysfunctional negative emotions (categories “sadness/depression” and “fear”).

The scoring is done directly, by summing up the answers to each item, allocating from 1 to 5 points as follows: not at all = 1, very little = 2, medium = 3, much = 4, very much = 5. 

The higher the total score, the higher the level of emotionally distress.

A score less than or equal to 28—very low level of distress

A score 29–39—low level of distress

A score between 40 and 56—average level of distress

A score between 57 and 86—high level of distress

A score equal to or greater than 87—very high level of distress

The scale allows both the calculation of an overall score of distress, as well as separate scores for “functional fear,” “dysfunctional fear,” “Sadness/functional depression”, and “sadness/dysfunctional depression”. In the present paper, we calculated an overall score of distress [15]. 

### 2.5. Evaluating the Cognitive-Emotional Coping Questionnaire (CERQ)

It is a scale that measures cognitive strategies that a person adopts to cope with stress or negative events. CERQ is a self-assessment questionnaire consisting of 36 items that measure nine strategies of coping: self-blame, acceptance, ruminating, positive refocusing, refocusing on planning, positive reassessment, perspective, catastrophe, and blaming others. It can be used to identify the general style of cognitive coping of a person, as well as his cognitive coping strategies after living through a specific event such as the COVID-19 pandemic. 

CERQ 1 self-blame is a cognitive coping strategy that involves the presence of thoughts according to which the entire responsibility for the lived situation belongs to oneself, the blame is attributed to themselves, and there are concerns about ideas that relate to mistakes that the person has made.

CERQ 2 Acceptance—referring to those thoughts due to which we resign ourselves to what has happened, and we accept the situation, thinking that it can no longer be changed, and that life will keep going.

CERQ 3 Rumination—referring to the situation in which we continuously think and/or are always overly concerned with the feelings and thoughts we associate with a negative event.

CERQ 4 Positive refocusing—occurring when we think of other, more pleasant things. Instead of us, we think of the negative event experienced.

CERQ 5 Refocus on planning occurs when we think about the steps we need to take, we follow, to deal with a negative event, or when we think of a plan through to change a situation.

CERQ 6 Positive reappraisal—occurring when we mentally associate a positive meaning with one negative event in terms of personal development, thinking that the game will make us stronger, looking for its positive aspects.

CERQ 7 Putting into perspective—referring to those thoughts that reduce the severity of events compared to other games and emphasizes that there are worse things in the world.

CERQ 8 Catastrophizing—occurring when we repeatedly think about how terrible an experienced event was and the fact that it is the cruelest/most terrible thing that could happen; that it was much worse than what has happened to others.

CERQ 9 Blaming others—occurring when we blame others for what has happened to us when we hold others responsible for what has happened and/or when we think about the mistakes that others have made in this regard.

Scoring and interpretation: Likert scale in 5 points, from 1 to 5, where 1 is “almost never” and 5 is “almost always”. The sum of the quotas of the items included in the scale is calculated (simple assembly), and we obtain the score of a subscale. The gross score is converted to a T-share based on existing standards. T scores less than or equal to 38 is a very low score’ a score between 39 and 43 is a low score; a score between 43 and 48 is below average; a score between 48 and 53 is an average score; a score between 53 and 58 is an above-average score; a score between 58 and 62 is a high score; and a score greater than or equal to 62 is a very high score.

All participants gave their informed consent for completing the questionnaires. 

According to the authors’ recommendations, all questionnaires were analyzed and interpreted by the same hospital clinical psychologist [12,13,14,15,16]. 

All the participants also completed a supplementary questionnaire made by study investigators. The pieces of information were obtained (about gender, age, profession, marital status, whom the person was living with during the pandemic, education level, smoking status, alcohol consumption, other comorbidities (including psychiatric conditions such as anxiety and depression), any other stressful event during this period besides COVID-19, and whether they were able to talk to anyone about what they were going through during this period.

Statistical analyses were performed using an IBM SPSS STATISTICS 25.0 application for windows, (IBM, Chicago, IL, USA). Median (25th percentiles; 75th percentiles) was calculated for both scores (ordinal variables) and quantitative variables that did not have a normal distribution. In the case of small samples (*n* < 30), the normal distribution was checked. If this condition was met, the comparison of two means was performed using a t-test for independent samples with equal or unequal variations depending on the result of the Levene test. If the condition was not met, the samples were compared using the Mann–Whitney U test. When comparing more than two media, the one-way Anova test or the Kruskal–Wallis test was used when the conditions of the Anova test were not met. A *p*-value < 0.05 (two-tailed) was taken to indicate statistical significance.

The study was approved by the Hospital Local Ethics Committee (no code number).

## 3. Results

Several questionnaires (140) were given to the medical staff. Ten persons refused to fill out the surveys, and 15 surveys were incomplete and could not be used. The overall response rate was 82%. One hundred and fifteen health care workers (doctors, junior doctors, nurses, hospital caregivers/caretakers) responded correctly to our questionnaires, 48 (41.8%) in the non-COVID-19 department, and 67 (58.2%) were in the COVID-19 department. The latter was 31 (46.2%) in the pulmonology ward and 36 (54.8%) in the intensive care unit. Most of them, 102, were women (88.7%) with a median age of 42 years old (with youngest being 25 years old and the eldest being 65 years old). The majority—70 (69.3%) out of 115—were married and living with their family. Half of the participants 50 out of 105, underwent other stressful events during the period (such as deceased parents). Participants’ demographic characteristics are shown in Table 1. 

### 3.1. The Brief Illness Perception Questionnaire (IPQ)

IPQ questionnaire assessed illness perceptions and was completed only by 67 participants, as the rest could not imagine themselves “being in that position” (having the disease). This could be interpreted as a high level of anxiety. They were so scared that they could not even conceive of having COVID-19. This is a coping method by which a person denies reality (the fact that he or she could get sick) because they are too scared of the consequences.

Total median score for non-COVID-19 personnel was 32.5 and for COVID-19 staff was 32, without significant difference. A higher score reflects a more threatening/catastrophic view of the disease. Both groups have ratings that reflect a perception of the disease as being “moderately” threatening (32 and 32.5 out of a maximum of 80). As in this period, any respiratory patient could be infected with SARS-CoV-2 considering disease symptoms, this result is somewhat understandable. Although they were not working in the COVID-19 department, they had as many chances of attending COVID-19 patients. We could even suppose that the risk was higher in the non-COVID-19 department as the respondents had to be incredibly careful, as they did not know whether they were treating a COVID-19 patient or not, whereas their colleagues had certainty.

On the other hand, both groups had the same protection equipment, which could have given them a feeling of safety. There are no significant differences in consequence score (IPQ 1), timeline score (IPQ 2), treatment control score (IPQ 4), identity score (IPQ 5), illness concern (IPQ 6), coherence score (IPQ 7), emotional representation (IPQ 8), or causal item (IPQ 9) between medical staff working in COVID-19 and non-COVID-19 departments. We found the only statistically significant difference in personal control score (IPQ 3) *p* =0.013, or the control that a person feels they have over the illness. The higher scores in the COVID-19 department could be explained by the fact that these respondents, because they were directly involved, were better informed about disease outcome and treatment possibilities. The disease at this stage was new and unknown, with unpredicted outcomes, sometimes even death, with no internationally accepted treatment recommendations. The medical staff dealing directly with COVID-19 patients had a better understanding of the reality that they did not have a lot of control over the illness. The non-COVID-19 respondents, on the other hand, would only have imagined being able to control the disease as their perception of the new condition was not a perfectly accurate one. 

There were no differences in disease perception IPQ 9, the item that ranks the three most important factors causing illness (see Table 2). All responses given by the patients were grouped in six categories: (1: civic responsibility, 2: epidemiological context, 3: incompliance with work rules, 4: preexisting medical conditions, 5: at-risk behaviors, and 6: physiological factors).

The most frequent causes for possible disease, mentioned in both the COVID-19 and in the non-COVID-19 group, were: at-risk behavior (75%) in both groups, and preexisting medical conditions (37.3% in the COVID-19 group vs. 31.3% in the non-COVID-19 group). 

Group 5, “at-risk behavior”, included: smoking, unhealthy food, lifestyle (tiredness due to overworking, burnout, night shifts, stress), and contact with infected surfaces. All the above causes are controllable factors, showing that the respondents believe that it is up to them to prevent the disease by avoiding certain risk factors. This could be a coping mechanism, as it is well known that when you are in control of something, anxiety is lower. 

A subgroup analysis of COVID-19 staff in an intensive care unit versus staff in a pulmonology ward found that there was a statistically significant difference in IPQ 1 (illness effect on life) (3 vs. 5, *p* = 0.004) and IPQ 7(degree of understanding of the illness) (6 vs. 8.5). IPQ 7 is a reverse-rating item; a higher score means a better understanding of the disease, resulting in a less catastrophic perception of the illness. The pulmonology ward staff better understood the condition and imagined that it would affect their life more severely than those in ICU. The latter, used to critical situations, believed that the disease would change their lifestyle. ICU staff, in general, have a notably different psychological profile. They have to be emotionally stable and have a high capacity to resist stress. They often face situations that require self-control, perspicacity, and the ability to make decisions rapidly in critical cases. They are used to action, and they theorize less. 

### 3.2. Perceived Stress Scale (PSS 10) 

There were no statistical differences between the two groups in stress-perception scores (*p* = 0.268). The mean score was 16 for COVID-19 staff and 15 for non-COVID-19 departments, representing a medium/high level of stress, indicating that in recent weeks one person has been exposed to many demanding situations, which they have faced with difficulty. This may have a clinical significance and is characteristic of people with stress problems. This is very plausible as in the first month when little was known about the disease, any patient with respiratory issues could be a potential COVID-19 patient. The level of perceived stressed correlated positively with disease perception in general (total IPQ) (*r* = 0.54, *p* < 0.001), with consequence scores (IPQ1), timeline scores (IPQ 2), and illness concern (IPQ 6 and 8), and correlated negatively with personal scores (IPQ3) and treatment scores (IPQ 4) in all participants. 

The more catastrophic the threat, the higher the stress level. The more staff thought that the disease was not under control and the treatment did not help, the higher the stress levels. Concerning coping mechanisms, PSS 10 correlated positively with rumination (CERQ 3) (*r* = 0.32, *p*< 0.001), and catastrophizing (CERQ 8) (*r* = 0.33, *p*< 0.001), with no significantly statistical difference in the two groups.

### 3.3. Profile of Emotional Distress (PDE) 

There was no statistical difference in the emotional distress profile between the two groups (*p* = 0.262). Forty-four participants in the non-COVID-19 group and 48 in the COVID-19 group had a medium level of emotional distress. The scale allows both the calculation of an overall score of distress, as well as separate scores for “functional fear,” “dysfunctional fear,” “sdness/functional depression,” and “sadness/dysfunctional depression. In the present study, we evaluated only the overall score of distress. 

Emotional cognitive coping evaluation questionnaire (CERQ) [16].

There were several coping mechanisms. In this paper, we focused on cognitive coping. Social support has two categories: instrumental and emotional. For this paper we evaluated, through the additional questionnaire, the emotional support. More than half of our staff, 59.5%, broadly received support from their loved ones: family, friends, or partners. 

Emotional cognitive coping evaluation did not differ significantly between the two groups, as both COVID-19 and non-COVID-19 staff were used to refocusing on planning (CERQ 5) and positive reappraisal (CERQ 6) as the usual methods of coping. However, COVID-19 medical staff used more positive refocusing (CERQ 4) than non-COVID-19 medical staff (*p* = 0.043). It was also noticed that patients that used self-blame (CERQ 1) as a coping modality did not use CERQ 4 (positive refocusing) and CERQ 6 (positive reappraisal) to cope with emotional distress. Respondents used positive refocusing (CERQ 4), a refocus on planning (CERQ 5), positive reappraisal (CERQ 6), and putting into perspective (CERQ 7) as coping methods more than the general population. Acceptance (CERQ 2) did not correlate with catastrophizing (CERQ 8) and blaming others (CERQ 9). Rumination (CERQ 3) did not correlate with acceptance (CERQ2) and positive refocusing ((CERQ 4). The latter did, however, correlate with refocusing on planning (CERQ 5), positive reappraisal (CERQ 6), and putting into perspective (CERQ 7). Catastrophizing (CERQ 8) correlated positively with blaming others (CERQ 9) and with those who have a more threatening view of the disease, imagining that they would have a severe form of the disease and that treatment would not help (IPQ 3). Profile of emotional distress—PDE—correlated positively with rumination (CERQ 3) (*r* = 0.33, *p* = 0.001) and catastrophizing (CERQ 8) (*r* = 0.39, *p* < 0.001) in all subjects. 

When evaluated separately, we noticed, in COVID-19 participants when considering their profession (doctor, junior doctor, nurse, health caregivers), a statistical significant difference only for: age (*p* < 0.001), self-blame (CERQ 1) (*p* = 0.019), treatment control (IPQ = 4) (*p* = 0.02), coherence(IPQ 7) (*p* = 0.045), and illness perception in general (IPQ total) (*p*= 0.039). The only clinically significant differences were in consequence scores (IPQ1) (*p* = 0.004) and in coherence scores (IPQ 7) (*p* = 0.006), with higher values in the pulmonology ward compared with the intensive care staff. We found no infected personal healthcare due to good circuits and protections. 

The fact that our staff did not use catastrophizing and blaming others as a coping mechanism explains the absence of an extremely high level of perceived stress and emotional distress. 

## 4. Discussion

Our study evaluated disease perceptions, perceived stress, emotional distress, and coping strategies in the first month of the COVID-19 pandemic among health care workers (doctors, junior doctors, nurses, caretakers) in a teaching pulmonology hospital from Cluj-Napoca, Romania. Although we would have expected a difference between the two groups, with a higher level of perceived stress and emotional distress among participants working in the COVID-19 department as they were facing a highly contagious, unknown disease, the difference between the two groups was not statistically different. One possible explanation for this might be that the staff in the COVID-19 department volunteered to be there, as the other responders refused to work there. Even in a subgroup analysis comparing intensive care unit staff with pulmonology wards, the difference remains insignificant. COVID-19, a highly infectious disease with a long incubation period and an unknown outcome that is lethal in some cases and without a specific treatment, is a stress source with significant influence, for both individuals and public social groups. It is only reasonable that it has a high impact on the public’s mental health [9]. In China, approximately 1700 healthcare personnel have been reported as infected with the COVID-19 disease (14.8% severely or critically ill, and 5% associated with deaths) [17]. Considering previous experience, we could assume that the COVID-19 outbreak will generate psychological reactions, which will lead to psychological disorders such as acute stress disorder, post-traumatic stress disorder, depression, and, in some cases, unfortunately, even suicide [4,6,7,8,9]. Different individuals may experience different levels of psychological reactions. SARS-CoV-2 viruses can be transmitted rapidly through respiratory droplets, contact, and aerosols, even by asymptomatic patients [9]. In China, for example, COVID-19 infection escalated quickly and without warning, with little time for emergency procedures or other mitigation efforts [17]. As they spend more time with patients, sometimes healthcare workers see them dying and the pain and even the fear seems to be greater among nurses than among doctors. This combined with stressful work, sleep deprivation, low freedom, heavy responsibility, and a high degree of cooperation, leads to a higher susceptibility towards the disease. In our study, nurses had a much higher score of illness perception (total IPQ) compared to doctors, but a lower score than junior doctors (31.5 vs. 27.3 vs. 43.5; *p* = 0.039) [9]. As a coping mechanism, they prefer positive reappraisal (CERQ 6). In 2004, one study reported that about 20% of the population suffered from emotional depression during SARS, and the incidence for nurses was as high as 45% in Toronto [18]. Folkman reported precious data focusing on psychological wellbeing and the coping processes in a crisis, as a complement to the traditional focus in both the medical and behavioral sciences on psychiatric symptoms [19]. Health care workers are on the front line of any given epidemic and risk their lives for their duty [8]. Health professionals have the responsibility of care to the infected patients, close contacts with patients families, and face the prospect of a public inquiry (3). In a recent study, Styra showed that caring for only one patient with SARS is significantly more stressful than caring for none or caring for two or more patients with SARS [20]. Liu H and colleagues showed that anxiety, fear, and anger are positively related to problem-focused coping and emotion-focused coping, meaning that there may be a phenomenon relating to “the more coping, the more panic” or “the more panic, the more coping”, or a “coping-panic cycle” [9].

We cope with success when we are exposed to a stressful situation if required to assess our self-efficacy, which entails self-appraisal of our ability to deal with the stressful event [20]. 

Most of the research on the psychological responses to infectious disease outbreaks have been conducted on either affected individuals, such as in the case of HIV/AIDS, or on health care workers working to combat an illness such as SARS [21].

An analysis of the direction of the relationship between emotion and coping found that problem-focused coping has a significant predictive effect on anxiety and sadness. The explanatory power of anxiety is 3.4%, and the explanatory power of sadness is 0.9%, indicating that the more problem-focused coping, the more anxious and sadder it can be. Although both problem-focused coping and emotion-focused coping have significant predictive effects on fear, the total explanatory power is 6%. Still, the explanatory power of problem-focused coping is 5.6%, and the explanatory power of emotion-focused coping is only 0.4%. Therefore, the more problem-focused coping, the more fear. Problem-focused coping and emotion-focused coping both have a significant predictive effect on anger, with a total explanatory power of 2.6%, and a problem-focused coping interpretation of 2.2%, but the explanatory power of emotion-focused coping is only 0.4%. It shows that problem-focused coping can predict emotional responses to a certain extent. It indicates that there may be a phenomenon that “the more problem-focused coping, the more anxious, the angrier, the more sadness [9].

On these scales, problem-focused coping strategies were rated higher (M = 3.1, SD = 0.57, *n* = 1242) than emotion-focused strategies (M = 2.6, SD = 0.63, *n* = 1242), concerning their relevance in dealing with the COVID-19 coronavirus disease. For emotion-focused coping, a significant difference between women and men can be found (*p* < 0.001). Women (M = 2.69, SD = 0.61, *n* = 630) agree more often with emotion-focused strategies than men (M = 2.52, SD = 0.65, *n* = 612). The same applies to problem-focused strategies. Women (M = 3.18, SD = 0.56, *n* = 612) agree more often with problem-focused strategies than men (M = 3.04, SD = 0.58, *n* = 612). Regarding respondents’ age, it was found that the older the respondent, the less likely they would be to use emotion-focused coping strategies (*r* = −0.14, *p* < 0.01). No significant correlations could be found for problem-focused strategies and age.

In detail, from all respondents (who answered either with “strongly agree” or “agree” on a 5-point Likert Scale, *n* = 1242), 87.2% wash or disinfect their hands more often than usual, 87.2% avoid public places/events, 85.2% are avoiding public transport (subway, tram, bus, train), 78.0% avoid contact with risk groups (older people, people with previous illnesses), 15.9% have bought more increased quantities of hand disinfectant/soap, or intend to buy more soon, 14.0% have bought increased quantities of staple foods (flour, sugar, pasta, rice, canned food) due to COVID-19 (coronavirus SARS-CoV-2), or will buy more shortly, and 6.8% have purchased large quantities of toilet paper and other hygiene items or will buy more quickly [22]. 

Older people estimate the risk of COVID-19 to be less than younger people. This can lead to political countermeasures such as social distancing being met with little acceptance and thus being less likely to be followed. Women are more concerned about infection of COVID-19 than men. People especially worry about being infected in places with a lot of public traffic, such as on public transport and in shops or restaurants. Coping strategies are highly problem-focused, and most of the respondents listen to expert advice and try to behave calmly and appropriately. People accept that COVID-19 will take time to tackle. Storing food is mainly justified by recourse to a combination of convenience and perception of the need to be prepared for a possible quarantine. 

## 5. Conclusions

In our study, there were no significant differences between the COVID-19 staff group and non-COVID-19 staff in terms of illness perception, stress perception, and emotional distress in the first month of disease. There are several explanations for this. As it was the first month and the beginning of the epidemic, with very little and often contradictory information, any respiratory patient could have been infected with COVID-19. After this period, doctors that went to work with SARS-CoV-2-infected patients were volunteers and were younger, and were therefore, at least in theory, less susceptible to be severely affected by of disease. Nevertheless, even if all personnel, no matter their department, had adequate protection equipment, it does not explain the high level of perceived distress. The most common coping mechanism was positive reappraisal and a refocus on planning, which seemed to be more prevalent than it was in the general population. Further research is required to see the consequences in the long term. 

## Figures and Tables

**Table 1 ijerph-17-04899-t001:** Participant characteristics.

Criteria	Total (*n* = 115)	Non-COVID-19 (*n* = 48)	COVID-19 (*n* = 67)	*p*-Value
**Male no. (%)**	13 (11.3)	4 (8.3)	9 (13.4)	0.394
**Age (Years Old) (*n* = 107) (%)**	40.78±9.58 42 (33;48)	43 (29;48)	42 (34;48)	0.950
**Marital Status (*n* = 101) (%)**				
married (%)	70 (69.3)	30 (68.2)	40 (70.2)	0.397
divorced (%)	6 (5.9)	1 (2.3)	5 (8.8)	
single (%)	13 (12.9)	7 (15.9)	6 (10.5)	
widow (%)	4 (4.0)	1 (2.3)	3 (5.3)	
in a relationship (%)	8 (7.9)	5 (11.4)	3 (5.3)	
**Living With (*n* = 101)**				
wife/husband + children (%)	50 (49.5)	23 (52.3)	27 (47.4)	0.776
just wife/husband (%)	20 (19.8)	9 (20.5)	11 (19.3)	
just parents (%)	5 (5)	2 (4.5)	3 (5.3)	
just children (%)	7 (6.9)	1 (2.3)	6 (10.5)	
family (sister, brother) (%)	3 (3.0)	1 (2.3)	2 (3.5)	
boyfriend/girlfriend (%)	9 (8.9)	5 (11.4)	4 (7)	
alone (%)	7 (6.9)	3 (6.8)	4 (7)	
**Alcohol (*n* = 96)**				
no (%)	20 (20.8)	10 (23.8)	10 (18.5)	0.527
On occasions (%)	76 (79.2)	32 (76.2)	44 (81.5)	
**Comorbidities (Others than Psychiatric Disorders). *n* = 68 (%)**	31 (27)	17 (54.8)	14 (37.8)	0.161
Psychiatric and physiological comorbidities (%)	22 (19.1)	8 (16.7)	14 (20.9)	0.570
Depression (%)	9 (7.8)	5 (10.4)	4 (6)	0.487
Anxiety (%)	14 (12.2)	5 (10.4)	9 (13.4)	0.626
Others stressful events (*n* = 47)				
Yes (%)	6 (12.8)	4 (22.2)	2 (6.9)	0.185
No (%)	41 (87.2)	14 (77.8)	27 (93.1)	
**Profession**				
Health caregiver (%)	23 (20)	9 (18.8)	14 (20.9)	0.478
Nurse (%)	46 (40)	19 (39.6)	27 (40.3)	
Junior doctor (%)	24 (20.9)	13 (27.1)	11 (16.4)	
Doctor (%)	22 (19.1)	7 (14.6)	15 (22.4)	
**School (*n* = 101)**				
secondary school (%)	7 (6.9)	2 (4.5)	5 (8.8)	0.627
high school (%)	18 (17.8)	8 (18.2)	10 (17.5)	
nurse college (%)	34 (33.7)	13 (29.5)	21 (36.8)	
university (%)	42 (41.6)	21 (47.7)	21 (36.8)	
**Smoking Status (*n* = 95)**				
Active smoker (%)	37 (39.4)	9 (21.4)	28 (53.8)	0.003
Former smoker (%)	8 (8.5)	3 (7.1)	5 (9.6)	
Non-smoker (%)	49 (52.1)	30 (71.4)	19 (36.5)	
**Support (*n* = 57)**	In text			
None (%)	1 (1.8)	1 (5.9)	0 (0)	0.183
to a small extent (%)	10 (17.5)	1 (5.9)	9 (22.5)	
Some way (%)	12 (21.1)	3 (17.6)	9 (22.5)	
Largely (%)	34 (59.6)	12 (70.6)	22 (55)	

**Table 2 ijerph-17-04899-t002:** IPQ 9 item in COVID-19 and non-COVID-19 groups.

Criteria	Total	Non-COVID-19	COVID-19	*p*-Value
IPQ 9 cod 1 (*n* = 56) no. (%) civic responsibility	7 (12.5)	3 (9.4)	4 (16.7)	0.447
IPQ 9 cod 2 (*n* = 56) no. (%) epidemiological context	16 (28.6)	9 (28.1)	7 (29.2)	0.932
IPQ9 cod 3 (*n* = 56) no. (%) incompliance with work rules	18 (32.1)	10 (31.3)	8 (33.3)	0.869
IPQ 9 cod 4 (*n* = 56) no. (%) preexisting medical conditions	19 (33.9)	10 (31.3)	9 (37.5)	0.625
IPQ 9 cod 5 (*n* = 56) no. (%) At-risk behaviors	42 (75.0)	24 (75)	18 (75)	1.000
IPQ 9 cod 6 (*n* = 56) no. (%) physiological factors	2 (3.6)	0 (0)	2 (8.3)	0.179

Question number 9 (IPQ 9) is the causal item, evaluating the three most important factors causing their illness.

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
