# Peer review of "Disease Perception and Coping with Emotional Distress During COVID-19 Pandemic: A Survey Among Medical Staff"

_ijerph, 2020, doi:10.3390/ijerph17134899_

Round 1

Reviewer 1 Report

The English needs much improvement.  For example,

abstract, line 19  suggest "pandemic as a global"

             line 22   suggest "The aims of our study were to"

             line 23   suggest "in a tertiary"

             lines 29-30   suggest "there were no differences between persons who worked in a COVID 19 department versus those working in a non-COVID 19 department."

              lines 31-32  suggest "As coping mechanisms, refocusing on planning and positive appraisals were used more than in the general population."

              line 34  suggest "patients versus those staff who were not handling  COVID patients in the first month"

Line 41  suggest "The World Health Organization (WHO)"

Line 170  Line was moved to line below, not necessary

219  Ten persons refused to

220   and 15 surveys wee incomplete

270   What is the heading for the far right column?

407   This paragraph needs to discuss the contrast between the perceptions of women and the old relative to actual vulnerabilities in which men and the old are far more vulnerable rather than less.

Author Response

please see the attachemnt 

Reviewer 2 Report

I enjoyed reading this research report. I was eager about the hypothesis stated at the end of page 2: that health workers working in COVOD-19 departments would be more stressed than staff in non-COVOD-19 departments in the Napoca, Romania hospital in which the researchers ran the study. However, despite a complex set of questionnaires administered the hypothesis was not carried. This study is likely to be an odd one out there compared to other studies which I believe would have similar hypotheses being borne out. It would therefore be of interest to the scholarly community. Maybe with time, the study can be rerun, perhaps at a different hospital in Romania.

Also I am not very convinced about the reason given for the hypothesis not being confirmed: that COVOD-19 was rather a new disease at the time the study was done.

We already knew how devastating it was in China before it arrived on the shores of Europe, and surely these health professionals were also following the news! The researchers might think harder for a better explanation.

I would have recommended to accept the paper as it is but I think there are a few writing infelicities that need to be addressed.

page 1. COVID-19 was 'described'....shouldnt it be: was discovered?

Page 2: change 'could infected' to could infect

Change 'the risk perceptions is not'...to: the risk perceptions ARE not....

Page 3:

Begin 2.2. with: This is a nine items questionnaire...

(These were just a few...but the authors should go through the work diligently to correct all writing infelicities!)
